# FLASH Radiotherapy and the Use of Radiation Dosimeters

**DOI:** 10.3390/cancers15153883

**Published:** 2023-07-30

**Authors:** Sarkar Siddique, Harry E. Ruda, James C. L. Chow

**Affiliations:** 1Department of Physics, Toronto Metropolitan University, Toronto, ON M5B 2K3, Canada; sarkar.siddique@torontomu.ca; 2Centre of Advance Nanotechnology, Faculty of Applied Science and Engineering, University of Toronto, Toronto, ON M5S 3E4, Canada; harry.ruda@utoronto.ca; 3Department of Materials Science and Engineering, University of Toronto, Toronto, ON M5S 3E4, Canada; 4Radiation Medicine Program, Princess Margaret Cancer Centre, University Health Network, Toronto, ON M5G 1X6, Canada; 5Department of Radiation Oncology, University of Toronto, Toronto, ON M5T 1P5, Canada

**Keywords:** FLASH radiotherapy, ultra-high dose rate radiotherapy, detectors, dosimetry

## Abstract

**Simple Summary:**

FLASH radiotherapy (RT) delivering ultra-high dose rate radiation can reduce normal tissue toxicity while effectively treating tumors. However, implementing FLASH RT in clinical settings faces challenges like limited depth penetration and complex treatment planning. Monte Carlo simulation is a valuable tool to optimize FLASH RT. Radiation detectors, including diamond detectors like microDiamond and ionization chambers, play a crucial role in accurately measuring dose delivery. Moreover, optically stimulated luminescence dosimeters and radiochromic films are used for validation. Advancements are being made to improve detector accuracy in FLASH RT. Further research is needed to refine treatment planning and detector performance for widespread FLASH RT implementation, which can potentially revolutionize cancer treatment.

**Abstract:**

Radiotherapy (RT) using ultra-high dose rate (UHDR) radiation, known as FLASH RT, has shown promising results in reducing normal tissue toxicity while maintaining tumor control. However, implementing FLASH RT in clinical settings presents technical challenges, including limited depth penetration and complex treatment planning. Monte Carlo (MC) simulation is a valuable tool for dose calculation in RT and has been investigated for optimizing FLASH RT. Various MC codes, such as EGSnrc, DOSXYZnrc, and Geant4, have been used to simulate dose distributions and optimize treatment plans. Accurate dosimetry is essential for FLASH RT, and radiation detectors play a crucial role in measuring dose delivery. Solid-state detectors, including diamond detectors such as microDiamond, have demonstrated linear responses and good agreement with reference detectors in UHDR and ultra-high dose per pulse (UHDPP) ranges. Ionization chambers are commonly used for dose measurement, and advancements have been made to address their response nonlinearities at UHDPP. Studies have proposed new calculation methods and empirical models for ion recombination in ionization chambers to improve their accuracy in FLASH RT. Additionally, strip-segmented ionization chamber arrays have shown potential for the experimental measurement of dose rate distribution in proton pencil beam scanning. Radiochromic films, such as Gafchromic^TM^ EBT3, have been used for absolute dose measurement and to validate MC simulation results in high-energy X-rays, triggering the FLASH effect. These films have been utilized to characterize ionization chambers and measure off-axis and depth dose distributions in FLASH RT. In conclusion, MC simulation provides accurate dose calculation and optimization for FLASH RT, while radiation detectors, including diamond detectors, ionization chambers, and radiochromic films, offer valuable tools for dosimetry in UHDR environments. Further research is needed to refine treatment planning techniques and improve detector performance to facilitate the widespread implementation of FLASH RT, potentially revolutionizing cancer treatment.

## 1. Introduction

While radiotherapy (RT) utilizes ionizing radiation to damage and eliminate cancer cells, radiation-induced toxicity restricts the maximum deliverable dose [1,2]. Ultra-high dose rate (UHDR) RT, known as FLASH RT, can solve this problem, as it delivers radiation at a rate several orders of magnitude higher than conventional clinical RT [3]. The FLASH effect, referred to as UHDR (≥40 Gy/s) RT, reduces damage to healthy tissues while maintaining antitumor effectiveness [4]. The flash effect, now termed FLASH RT, was initially reported by Dewey and Boag in 1959, but gained prominence after 2014 with in vivo studies demonstrating reduced normal tissue toxicity while achieving similar tumor control compared to conventional RT [5].

The first patient treated with FLASH RT was a 75-year-old individual with multiresistant CD30+ T-cell cutaneous lymphoma that had disseminated throughout the skin surface. FLASH treatment was delivered using a specialized 5.6 MeV LINAC, designed specifically for FLASH RT. The prescribed dose to the planning target volume was 15 Gy delivered in 90 milliseconds (ms). Dosimetric measurements using GafChromic films and alanine were performed to ensure dose consistency [6].

Numerous in vivo studies have investigated the FLASH effect and its potential benefits. For instance, a study evaluated lung fibrogenesis in mice subjected to UHDR irradiation and conventional dose rate irradiation, demonstrating improved outcomes and spared normal smooth muscles and epithelial cells from acute radiation-induced apoptosis with UHDR irradiation [7]. FLASH RT shows promise as a treatment option with significant potential for improving outcomes, particularly for pancreatic cancer, which currently faces limitations due to gastrointestinal toxicity [8].

However, the clinical implementation of FLASH RT presents technical challenges. Conventional linear accelerators are unable to generate therapeutic doses beyond a 15 cm depth, limiting FLASH RT to skin cancers or tumors located close to the body surface [9]. Treatment planning for FLASH RT is complex and currently under investigation to determine the best methods and optimization techniques. Several studies have explored the application of Monte Carlo (MC) codes for dose calculation in FLASH RT [10]. Moreover, dosimetry in FLASH RT is challenging due to the delivery of high instantaneous doses, necessitating a comprehensive understanding of factors influencing detector response [11].

Although FLASH RT has primarily been studied using X-rays, the FLASH effect has been validated in preclinical experiments using electrons and protons, with both particle types operated at mean dose rates above 40 Gy/s [12]. Notably, the immunological memory response in mice was found to be similar between electron and proton beams, independent of dose rate [13]. Fractional delivery in FLASH RT typically involves a sequence of pulses with a frequency of approximately 100 Hz (interval between pulses ≈ 10 ms) and a dose per pulse greater than 1 Gy, enabling fraction delivery within a few tenths of a second [14].

The sparing effect of FLASH RT on normal cells is influenced by oxygen depletion, with varying oxygen levels in tumors and normal tissues affecting the efficacy of the FLASH effect [15]. Determining the precise dose required to induce the effect is crucial and requires further investigation [16]. Studies using carbon ion irradiation explored the response of CHO-K1 cells to irradiation at different dose rates under various levels of oxygenation. FLASH irradiation with a dose rate of 70 Gy/s demonstrated a significant FLASH effect and oxygenation dependence [17]. Furthermore, FLASH RT has been found to spare normal tissue temporarily due to hypoxia resulting from oxygen depletion induced by UHDR irradiation [18]. Depleting cellular oxygen at the FLASH dose rate was shown to be achievable with an oxygen concentration of 0.4% and a dose rate of 5–10 Gy [19]. Dosimetry performance and optimization of FLASH dose rates have been systematically evaluated in hypofractionated lung cancer patients, enabling the optimization of Bragg Peak and transmission plans to achieve acceptable plan quality [19].

Moreover, FLASH irradiation induces different cell death mechanisms, including pyroptosis, apoptosis, and necrosis, with varying ratios in cancer stem cells and normal cancer cells. Cancer stem cells exhibit greater resistance to radiation under FLASH irradiation, potentially due to increased lysosome-mediated autophagy and decreased necrosis, apoptosis, and pyroptosis. Further investigations are warranted to better understand the radioresistance of cancer stem cells [20].

## 2. Monte Carlo Simulation

MC simulation is recognized as one of the most accurate methods for dose calculation in RT [21]. Bazalova-Carter et al. investigated the application of MC methods in percentage depth dose calculation using electron beams of different sizes (50 and 70 MeV). The EGSnrc/BEAMnrc and DOSXYZnrc MC codes were employed to calculate the dose in a polystyrene phantom. The simulation results exhibited good agreement (within 5%) with the measured data for depth–dose curves and beam profiles. However, there was a discrepancy of 42% between the calculated and measured doses [21].

Palma et al. utilized the same MC codes (EGSnrc/BEAMnrc and DOSXYZnrc [22,23]) to perform dose distribution calculations for very high-energy electron beams in five clinical cases. Additionally, MC simulation was employed for dose calculation using two 160 kV X-ray tubes, where the difference between experimental results and simulations was within 3.6% [24]. Geant4 is another widely used software for simulating particle transport in matter and has been employed for dose calculation and new hardware design in FLASH research [24]. In another study, BEAMnrc MC codes were used to model a LINAC. The resulting phase-space file from the simulation was fed into DOSXYZnrc to calculate the 3D dose distribution in a voxel-based phantom. Comparison between the simulated and experimentally measured results showed good agreement for different maximum dose ranges (R_max_, R_90_, R_80_, and R_50_). The deviation between the MC-calculated percent depth dose (PDD) curves and the measurements was 5.2% [25]. EGSnrc (release v2023) MC software modules, namely BEAMnrc and DOSXYZnrc, were employed to create a treatment plan for whole-brain RT. The simulation demonstrated that two lateral opposing 40 MeV electron beams could be used to deliver a FLASH dose rate of >115 Gy/s for whole-brain RT, highlighting its potential for clinical application [26].

The UHDR of FLASH therapy presents new challenges, such as the need for a new shielding system. MC simulation can provide a solution for simulating such a shielding system, as explored in a study [27]. Another investigation focused on ionizing radiation acoustic imaging through simulation and its potential as a dosimetric tool for FLASH RT. Ionizing radiation acoustic imaging is an imaging technique that creates dose-related images by utilizing acoustic waves generated through the thermoacoustic effect in response to ionizing radiation. A full 3D dose distribution was simulated using the EGSnrc (BEAMnrc/DOSXYZnrc [22,23]) MC simulation code in a phantom with a 1 × 1 cm^2^ field. The simulation results were verified using Gafchromic films. The experimental measurements and dose simulation agreed within an approximately 5% relative error for the central beam region at up to 80% dose, both for the central profile region and the percentage depth dose. This study demonstrates the feasibility of utilizing ionizing radiation acoustic imaging as a dosimeter for depth–dose measurement and beam localization in FLASH RT [28]. These studies highlight the use of MC simulation in dose calculation, treatment planning, hardware design, shielding system simulation, and dosimetry for various aspects of RT, including both conventional and FLASH techniques.

## 3. Radiation Dose Detectors

Radiation detectors play a vital role in various fields, including medical physics, radiation protection, and high-energy sensitive imaging [29]. Recent advancements in detector technology have introduced exciting developments, such as the photon counting detector that utilizes semiconductor materials to generate electronic signals in response to incident X-ray photons [30]. Another noteworthy innovation is the pressurized ionization chamber detector, which enables the characterization of alpha and beta radioactive sources and can measure radioactive sources in internal 2π or 4π geometry [31]. Furthermore, the availability of 2D and 3D ionization chamber arrays allows for real-time dose verification [32]. The application of these novel dosimetric technologies in UHDR dose delivery holds significant promise.

In the context of FLASH RT, UHDR per pulse is necessary to achieve the FLASH effect. However, real-time dosimetry poses a significant challenge. Conventional vented ionization chambers used for dosimetry exhibit substantial deviation from linearity as the dose rate per pulse increases, primarily due to recombination losses in the sensitive air volume. Solid-state detectors offer good response stability with respect to accumulated dose and present a promising alternative. Diamond detectors, among other solid-state detectors, have been extensively utilized in RT applications [33].

These advancements in radiation detector technology enhance the accuracy and reliability of dose measurements, allowing for improved outcomes in various fields, including FLASH RT and conventional RT.

### 3.1. Diamond Detector

Diamond detectors possess high radiosensitivity and offer excellent spatial resolution, making them well suited for applications involving large dose gradients and small fields [34]. The viability of diamond detectors, such as microDiamond, is being investigated for their potential use in FLASH RT. The microDiamond detector functions as a Schottky diode, where the sensitive volume of a diamond is positioned between a metallic contact and a p-type diamond structure that serves as the back contact. This arrangement generates a depletion region in the contact area, which possesses an inherent potential and serves as the sensitive volume. As a result, there is no need for an external bias voltage to operate the detector. Figure 1 illustrates the equivalent circuit of this diode [35].

In the context of FLASH RT, a specific diamond detector designed for dosimetry purposes was introduced. The study focused on its application in both ultra-high dose per pulse (UHDPP) and UHDR beams utilized in FLASH RT. The detector was successfully implemented in an electron FLASH LINAC, and it exhibited linearity within the dose per pulse range. The study demonstrated strong agreement between dose per pulse, output factor (ratio of the dose in air for a given field to that for a reference field), and beam profile measurements when compared to a reference detector [36].

To address the inherent response nonlinearities observed in conventional detectors, a novel diamond-based Schottky diode detector was developed. The prototype’s response linearity was influenced by the size of its active volume and series resistance. However, through proper tuning and adjustment, the detector layout was able to achieve linearity up to at least 20 Gy/pulse [37].

The unique properties of diamond detectors, along with their improved linearity and dose measurement capabilities, make them promising candidates for enhancing dosimetry accuracy in both FLASH RT and conventional RT settings.

### 3.2. Ionization Chamber

In clinical practice, ionization chambers are commonly used for both absolute and relative dose measurements in radiation therapy. These chambers are particularly useful in regions with high dose gradients [38]. For FLASH RT, specific ionization chambers such as the 2D strip segmented ionization chamber array were developed for the experimental measurement of 2D dose rate distributions [39]. However, the standard ionization chamber can be significantly affected by UHDR per pulse due to the electric field generated by the large density of charges from the dose pulse [40].

To address the dosimetric challenges associated with UHDR per-pulse irradiation, researchers have explored modifications and calculation methods for ionization chambers. A study introduced a new calculation method for the free electron fraction in an ionization chamber. By modeling the capture process of electrons and evaluating the free electron fraction, they were able to estimate the response of the ionization chamber after irradiation [41]. Another study proposed an empirical model of ion recombination in an ionization chamber for UHDR per pulse electron beams. The study compared the observed ion recombination output with various theoretical models and found that taking ion recombination into account enables the ionization chamber to function for dose measurements at UHDR per pulse [42].

In the context of proton therapy and FLASH irradiation, different models of ionization chambers have been evaluated. One study investigated the response of four ionization chamber models for spread-out Bragg peak proton FLASH irradiation. The study found that plane-parallel chambers with smaller gaps between electrodes are more favorable for FLASH RT dose measurements [43]. Furthermore, efforts have been made to improve the ion collection efficiency of ionization chambers to make them suitable for FLASH RT. For example, the ion collection efficiency of vented ionization chambers was studied for the UHDR electron beam, and the dependences of the sensitive air volume on the design of chamber and electric field were evaluated. The results indicated a decrease in ion collection efficiency within the UHDR range. The extent of the decrease varied depending on factors such as electrode distribution, electric field strength, and chamber voltage in the sensitive air volume [44]. Another study developed and characterized an ultra-thin parallel plate ionization chamber that showed potential for extending the dose rate operating range to the ultra-high dose per pulse range used in FLASH RT. To accommodate the ultra-thin ionization chamber (UTIC) and a specifically modified diamond detector (referred to as flash-diamond) for UHDR, a polymethyl methacrylate (PMMA) phantom was constructed. The flash-diamond served as a reference dosimeter for the experiment, as shown in Figure 2 [45].

Additionally, novel ionization chamber technologies have been explored for online dosimetry in FLASH RT. The RazorTM Nano Chamber, with its small sensitive volume, has demonstrated higher ion collection efficiency compared to larger chambers, making it a potential tool for online dosimetry in FLASH RT [46].

The development and refinement of specialized ionization chambers, calculation methods, and online dosimetry tools are essential for advancing the field of radiation dosimetry in FLASH RT. These advancements aim to ensure accurate and reliable dose measurements in the context of UHDR delivery, facilitating the safe and effective implementation of FLASH RT in clinical practice.

### 3.3. Radiochromic Film

Radiochromic film is a dosimeter that possesses desirable characteristics for radiation responses, such as independence from radiation energy and dose rate, as well as a negligible volume effect [47]. The effectiveness of the different types of radiochromic film depends on their dose sensitivity, accuracy, and response to environmental conditions [48].

In the context of FLASH RT a study utilized Gafchromic^TM^ EBT3 radiochromic film to measure the dose in high-energy X-rays capable of triggering the FLASH effect in mice. The film was placed between the mice and the PMMA holder to measure the dose, and it was also used to validate MC simulation results [49]. Another investigation performed a dosimetric characterization of a plane-parallel ionization chamber under UHDR conditions using radiochromic films. Radiochromic films were used to verify the beamline setup, measure depth–dose distribution and dose profile, and serve as a reference for ionization chamber characterization. The study revealed significant recombination losses and polarity effects in the ionization chamber [50].

An electron-scattering device was created for the practical use of UHDR electron beams in FLASH preclinical research at the Dongnam Institute of Radiological and Medical Sciences [51]. The scattering device’s geometry for a 6-MeV linear accelerator was determined using Monte Carlo N-particle transport simulations. Radiochromic films were used to measure the off-axis and depth dose distributions with the scattering device. The measured dose per pulse varied from 4.0 to 0.2 Gy/pulse at different source-to-surface distances (SSD) ranging from 20 cm to 90 cm. At an SSD of 30 cm and a repetition rate of 100 Hz, the dose rate reached 180 Gy/s, providing a sufficient dose rate for conducting small-animal FLASH studies.

Furthermore, radiochromic film has been employed in various applications within the establishment of FLASH RT. In a study involving canine cancer patients, radiochromic film (GafChromic EBT-XD) was utilized for dose measurements on a phantom and to measure dose per pulse. The film was also used for in vivo dose measurements at the skin’s surface to verify the delivered dose. The experimental configuration depicted in Figure 3 illustrates the setup utilized for conducting measurements using radiochromic film to determine both the total dose and dose per pulse [52]. These measurements were correlated with the signal obtained from a Farmer-type ionization chamber (NE 2505/3-3A), which was positioned within a specially designed holder placed in the applicator.

In addition, radiochromic film was used in conjunction with the MC FLUKA code to measure dose in FLASH irradiation and investigate the enhancement of radio-resistance in normal fibroblast cells under conditions of hypoxia and mitochondrial dysfunction [53].

In proton FLASH dosimetry, different radiochromic films have been compared for their dose rate dependency. A study conducted at the ARRONAX cyclotron facility evaluated GAFchromic™ EBT-XD, GAFchromic™ EBT3, and OrthoChromic OC-1 films after proton irradiation. The study found that OC-1 films exhibited dose rate independence in proton beams up to 7500 Gy/s, while caution should be exercised when using EBT-XD and EBT3 films at dose rates exceeding 10 Gy [54]. Another study focused on dosimetry in proton pencil beam scanning FLASH RT, employing MC codes for simulations and Gafchromic^®^ EBT3 films for dose measurements. The investigation aimed to determine the absolute dose for FLASH proton beam radiotherapy using a primary standard proton calorimeter, achieving an uncertainty of 0.9% through the application of correction factors [55].

By leveraging the capabilities of radiochromic film and its compatibility with various dosimetric techniques and simulations, researchers continue to advance the field of dosimetry in FLASH RT, enabling accurate and precise dose measurements necessary for the safe and effective implementation of this promising treatment modality.

### 3.4. Alanine

Alanine dosimetry is a widely used method in high-dose dosimetry, relying on irradiated crystalline alanine that is measured using electron paramagnetic resonance (EPR) spectrometry. It is renowned for its exceptional stability in post-irradiation response [56]. Alanine dosimeters are commonly employed for calibration services and are suitable for a wide range of industrial applications due to their energy independence (above 100 keV) and minimal dose rate effects [57].

While alanine dosimetry is accurate, its application in FLASH RT for biological experiments and clinical use requires a reduction in reading time. One study focused on optimizing an alanine dosimeter by improving the acquisition of EPR spectra using a Bruker spectrometer. Parameters such as the number of scans, time constraints, conversion time, microwave power, and modulation amplitude of the magnetic field were investigated for optimization purposes [58].

In the context of specific radiation sources, another study compared an alanine detector with a PTW PinPoint ionization chamber (used as a reference) for an orthovoltage X-ray source with an average dose rate of 11.6 kGy/s. The study concluded that the alanine dosimeter is suitable for the UHDR calibration of orthovoltage X-ray sources [59]. Elsewhere, a study examined the use of an alanine-based dosimetry system to precisely evaluate absorbed dose to water in UHDR per pulse electron beams. The electron beam used in the study had a range of 0.15–6.2 Gy/pulse, and MC simulation was employed to calculate the conversion factor required for alanine dosimetry and determine the beam quality [60]. The absolute dosimetry of the Oriatron eRT6 linear accelerator was examined using alanine, thermo-luminescent dosimeters (TLD), radiochromic films, and an ionization chamber for relative stability [61]. A comparison of results between alanine, films, and TLD demonstrated a dose agreement within 3% for dose rates ranging from 0.078 Gy/s to 1050 Gy/s. This indicates that such dosimeters are suitable for absolute dosimetry in FLASH RT. A comparison was made between the reference dosimetry using the PinPoint ionization chamber and alanine dosimetry for synchrotron X-ray sources [59]. The results revealed a relative response of 0.932 ± 0.027 (1σ) for the alanine pellets irradiated at the European Synchrotron Radiation Facility (ESRF) compared to the ^60^Co facility at National Centre for Radiation Research and Technology. These findings took into account corrections for the ESRF polychromatic spectrum and the different field sizes used. Therefore, it can be confirmed that alanine is a suitable dosimeter for calibrating orthovoltage X-ray sources operating at UHDR.

Furthermore, research has demonstrated the applicability of alanine dosimeters and TLDs for dosimetry in FLASH RT. By imposing specific requirements on the procedure, such as optimizing conversion time, time constant, microwave power, modulation amplitude of the magnetic field, and the number of scans, a maximum dose deviation of 1.8% was achieved for the dose range of 10 Gy–100 Gy, while keeping the deviation to the reference within ±2% [58]. Moreover, studies have shown that alanine dosimeters exhibit good agreement with TLDs, and alanine dosimetry provides the closest match between the expected and measured doses. Figure 4 presents the bias of various detectors, including alanine, film, and TLD, in relation to the expected doses in both conventional RT and UHDR RT [62].

For the irradiation of biological models with pulsed electron beams at UHDR, dosimetry procedures involving alanine dosimeters, films, and TLDs have been investigated. These methods demonstrated dose agreements within 3% for dose rates ranging from 0.078 Gy/s to 1050 Gy/s, making them suitable for FLASH RT. The studies also emphasized the importance of appropriate setup and correction factors, as active dosimetry without them can lead to dose deviations of up to 15% of the prescribed dose. However, by following the proposed study setup and procedure, the deviations can be reduced to less than 3% [61].

The ongoing research and optimization efforts in alanine dosimetry highlight its potential for accurate and reliable dose measurements in the context of FLASH RT, paving the way for its integration into clinical practice and biological studies.

### 3.5. Radioluminescence, Cherenkov Radiation Dosimetry, and Others

Recently, there have been significant advancements in utilizing Cherenkov energy as a monitoring tool for biological changes, such as oxygen levels, during radiotherapy. Cherenkov emission occurs naturally as a byproduct of RT when high-energy charged particles surpass the local phase velocity of light within a dielectric medium, resulting in the emission of optical photons [63].

Studies have explored spatial-temporal beam profiling for electrons in UHDR conditions using Cherenkov emission, radioluminescence imaging, and complementary metal oxide semiconductor (CMOS) cameras. Surface dosimetry was investigated by imaging scintillation or Cherenkov emission from a solid water phantom (Gd_2_O_2_S:Tb) and comparing the optical imaging results with the response measured by Gafchromic film at various depths. The pulse-per-beam output from Cherenkov imaging agreed within 3% with photomultiplier tube Cherenkov output. Scintillation and Cherenkov emission showed linearity with dose (R^2^ = 0.995 and 0.987, respectively) and were independent of dose rate in the range of approximately 50 Gy/s to 300 Gy/s (0.18–0.91 Gy/pulse) [64].

In another study, a nitrogen-doped, silica-based multimodal optical fiber was examined for monitoring very UHDR conditions through radiation-induced emission. The findings indicated that the emission of radiation from this fiber exhibited a linear dependence on the dose rate over a broad range of dose rates (10^–2^ Gy(SiO_2_)/s to a few 10^9^ Gy(SiO_2_)/s) and photon energies (40 keV to 19 MeV). This is depicted in Figure 5, highlighting its significant potential for beam monitoring in UHDR scenarios [65].

Fricke or ferrous ammonium sulphate detectors, which are chemical-based dosimeters, rely on the oxidation of ferrous and ferric ions, followed by their interaction with ionizing radiation. These dosimeters possess properties similar to water since they consist of 96% water by weight. They can serve as absorbed doses to water primary standards in high-energy electron beams [66].

A novel plastic scintillator capable of resolving individual pulses with a temporal resolution as short as 2.5 ms was investigated in a study. The plastic scintillator’s response measurement exhibited linearity with ionization chamber measurement (within ≤1%) over a dose range of 4–20 Gy and pulse frequencies of 18–180 Hz. Under reference conditions, the plastic scintillator maintained its dose–response even under ultra-high pulsed dose rate conditions and agreed with EBT-XD film dose measurements within >4%. It demonstrated a linear and reproducible response, accurately measuring the absorbed dose from a 16 MeV electron beam with an ultrahigh pulsed dose rate [67].

One study focused on the first characterization of six real-time point scintillation dosimeters using five phosphors (Al_2_O_3_:C, Mg; Y_2_O_3_:Eu; Al_2_O_3_:C; (C_38_H_34_P_2_)MnBr_4_ and (C_38_H_34_P_2_)MnCl_4_) in an ultra-high pulsed dose rate electron beam. The linearity of response with dose was tested by varying the number of pulses, and a linearity with R2 > 0.9989 was observed up to at least 200 Gy [68].

The response of three detectors, Gafchromic EBT-XD film, optically stimulated luminescence (OSL), and the CC13 ionization chamber was investigated in UHDR conditions. Experimental results showed that EBT-XD film can be used in FLASH experiments without requiring any dose rate correction up to at least 2 × 10^4^ Gy/s. The agreement between the doses measured with film at different distances from the scattering foil and the doses computed using the effective inverse square law confirmed this. OSL measurements also exhibited agreement with the inverse square law, maintaining independence up to 280 Gy/s. The ionization chamber achieved reasonable agreement between the modeled and measured chamber efficiency; however, the discrepancies exceeded the clinically used tolerance of 2% [69]. Over the years, OSL has emerged as a strong competitor for thermoluminescence dosimetry and other dosimetry systems [70]. In spite of the promise that OSL offers in terms of UHDR conditions, it is limited by available materials, many of which (e.g., Al_2_O_3_:C) were first introduced in the 1960′s. The key will be to identify new materials specifically designed for FLASH—that is, with tuned bandgap, radiation hardness, high radiative recombination efficiency of trapped carriers, linearity in deep state creation as a function of dose up to high doses, and excellent minority carrier lifetime and transport. It is likely that such materials will rely on nanostructured materials where size quantization of electronic states can allow for tailored spectral output, enhanced exciton binding energies, and polarization anisotropy to provide for higher performance and more versatile materials and likely the next generation of OSL for FLASH.

The potential of lead-doped scintillator dosimeters for use in FLASH-capable UHDR X-ray beams was investigated. The study demonstrated that the lead-doped scintillators were independent of dose rate for UHDR X-rays in the range of 1.1 Gy/s to 40.1 Gy/s. When compared with MC simulations, the dose to water measured with the lead-doped (5%) scintillator detector agreed within 0.6% [71].

In the first positron emission tomography imaging and dosimetry study of a FLASH proton beam, the radiation environment was characterized using cadmium-zinc-telluride and a plastic scintillator counter [72].

A fiber optic radiation sensor created with a plastic scintillator, an optical filter, and a plastic optical fiber was explored for use in FLASH RT. The sensor detected radiation-induced emissions such as fluorescence and Cherenkov radiation generated within the transmitting optical fiber. The sensor’s output was measured at different distances from an electron scattering device and compared with the output of an ionization chamber and radiochromic films [73].

The EDGE detector, based on diodes, was also studied to characterize FLASH beams and its response compared to other detectors. The EDGE detector showed agreement with film measurements within 2% on average over the measured range of varying doses (up to 70 Gy), dose per pulse (up to 0.63 Gy/pulse), and dose rate (nearly 200 Gy/s). It also agreed with the W1 scintillation detector for dose per pulse (up to 0.78 Gy/pulse) within 2% on average. The EDGE detector demonstrated the ability to quantify the beam spatially and temporally with sub-millisecond resolution, making it suitable for in vivo studies [74].

These studies contribute to the advancement of dosimetry methods for FLASH RT and provide valuable insights into the performance and potential applications of various detectors in ultra-high dose rate scenarios.

## 4. Future Prospective

The future prospects of radiation dosimetry in FLASH RT hold significant promise for advancing this emerging treatment modality. Dosimetry plays a crucial role in accurately measuring and monitoring the dose delivered during RT, and its importance is further magnified in the context of FLASH RT, which involves ultra-high dose rates and unique delivery techniques.

One of the key areas of focus for future dosimetry in FLASH RT is the development of specialized detectors capable of accurately measuring the high dose rates associated with this treatment. Conventional dosimeters may exhibit limitations in their response time and saturation effects at such extreme dose rates. Research efforts are underway to explore novel dosimetry technologies that can provide real-time measurements and maintain accuracy in the presence of rapid dose delivery.

Additionally, there is a need to investigate the dosimetric properties of various radiation modalities used in FLASH RT, including electron beams, proton beams, and X-rays. Each modality may have distinct dosimetric characteristics, and understanding their behavior in the context of FLASH RT is crucial for optimizing treatment planning and ensuring accurate dose delivery. Comparative studies and advancements in MC simulation techniques can contribute to a deeper understanding of the dosimetric aspects specific to FLASH RT.

The development and validation of comprehensive dosimetry systems specifically designed for FLASH RT are also anticipated in the future. These systems would encompass not only dose measurement devices, but also data acquisition, analysis, and quality assurance tools tailored to the unique requirements of FLASH RT. Such systems would facilitate precise and reliable dose calculations, treatment verification, and patient safety in clinical implementations of FLASH RT.

Moreover, the exploration of advanced imaging techniques integrated with dosimetry in FLASH RT holds great potential. Real-time imaging modalities, such as in vivo dosimetry using electronic portal imaging devices or onboard imaging systems, can provide valuable information on dose distribution during treatment delivery. Combining imaging data with dosimetric measurements can enable continuous monitoring and adaptive strategies to further enhance the accuracy and safety of FLASH RT.

The future prospects of radiation dosimetry in FLASH RT are centered around the development of specialized detectors, comprehensive dosimetry systems, and integration with advanced imaging technologies. Continued research and collaboration between radiation oncologists, medical physicists, and engineers is vital to address the dosimetric challenges and unlock the full potential of FLASH RT as an innovative and effective cancer treatment option.

## 5. Conclusions

FLASH RT demonstrates tremendous potential as a cancer treatment option; however, further investigation is needed before it can be widely adopted. Future FLASH devices may require the ability to perform multiple-field conformal radiation to reduce toxicity in healthy tissues compared to single-field approaches [2]. While most of the current FLASH studies have focused on electron beams, proton beams, and X-ray beams, they have shown beneficial effects [75]. In addition to the dosimetry challenges associated with FLASH RT, further research is required for its successful clinical implementation [76]. Caution should be exercised during the clinical application of FLASH RT until a comprehensive understanding of the biological effects and a thoroughly tested dosimetry system are established [77]. Ongoing research endeavors to unravel the fundamental mechanisms responsible for the distinctive tissue-sparing benefits of FLASH radiation. By comprehending how FLASH RT influences biological responses in both healthy tissues and cancer cells, researchers hope to develop enhanced treatment protocols to enhance patient outcomes.

In particular, the FLASH effect is influenced by various factors such as total dose, dose rate, pulse rate, radiation modality, and fractionation. Hence, accurate dose monitoring is vital in delivering the desired effect. Continued research and investigation into suitable dosimeters for FLASH RT will facilitate its further development and implementation in diverse types of cancer treatments [78,79].

## Figures and Tables

**Figure 1 cancers-15-03883-f001:**
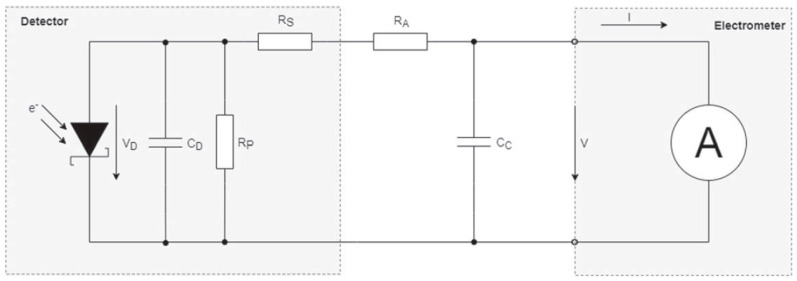
Circuit diagram for a diode representation of a diamond detector. Reproduced from reference [35] under the Creative Commons Attribution 4.0 International License (https://creativecommons.org/licenses/by/4.0/ (accessed on 1 July 2023)).

**Figure 2 cancers-15-03883-f002:**
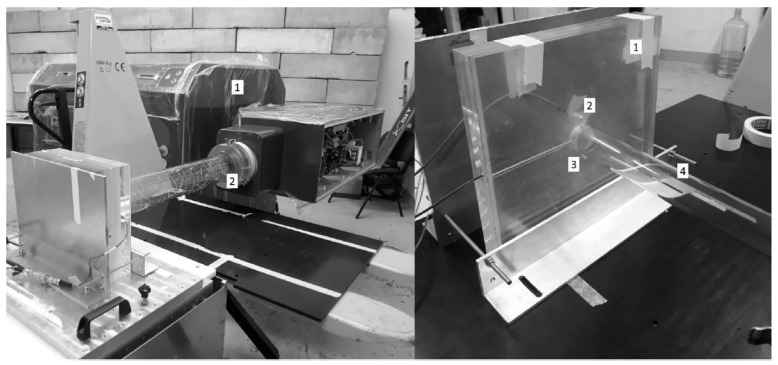
The experimental setup at SIT (Italy). On the left side, there is the ElectronFlash, a linear accelerator (1), which is utilized with a 100 mm diameter applicator (2). On the right side is the PMMA phantom (1) accompanied by the flash-diamond (2) and the ultra-thin ionization chamber (3) that are prepared for irradiation with the 35 mm diameter applicator (4). Reproduced from reference [45] under the Creative Commons Attribution 4.0 International License (https://creativecommons.org/licenses/by/4.0/ (accessed on 1 July 2023)).

**Figure 3 cancers-15-03883-f003:**
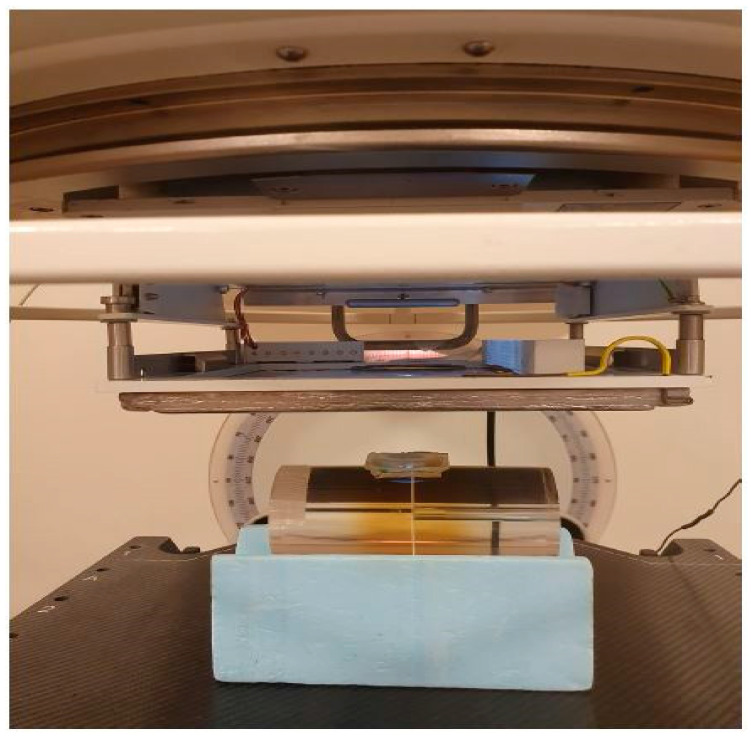
Experimental arrangement employed in the preparation of each patient’s treatment. Radiochromic film was utilized to conduct measurements on phantoms that simulated the treatment geometry. These measurements encompassed the total dose, number of pulses, and dose per pulse intended for delivery to the patients. A Farmer-type ionization chamber was employed as the output monitor. Reproduced from reference [52] under the Creative Commons Attribution 4.0 International Li-cense (https://creativecommons.org/licenses/by/4.0/ (accessed on 1 July 2023)).

**Figure 4 cancers-15-03883-f004:**
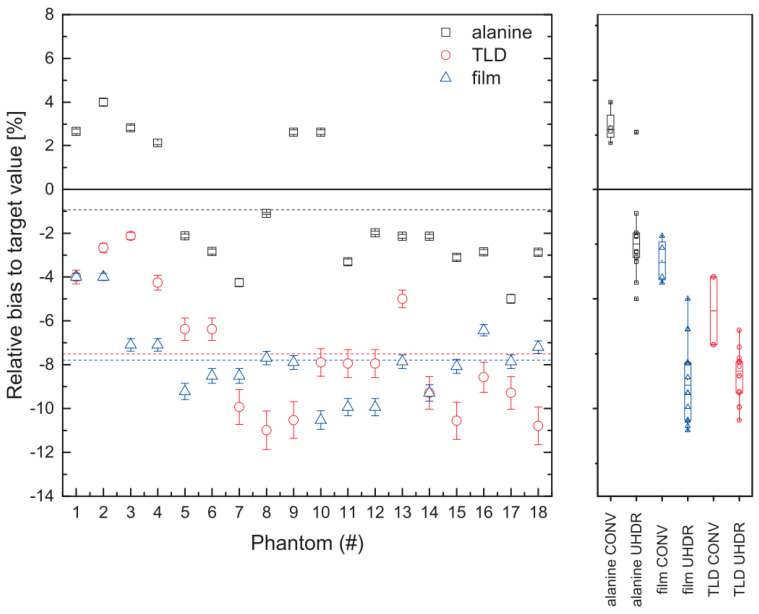
The (**left**) plot shows the relative bias to target value in phantom for alanine, film and TLD in conventional RT and UHDR RT. The (**right**) plot shows the biases for each dose detector and dose rate mode. Reproduced from reference [62] under the Creative Commons Attribution 4.0 International License (https://creativecommons.org/licenses/by/4.0/ (accessed on 1 July 2023)).

**Figure 5 cancers-15-03883-f005:**
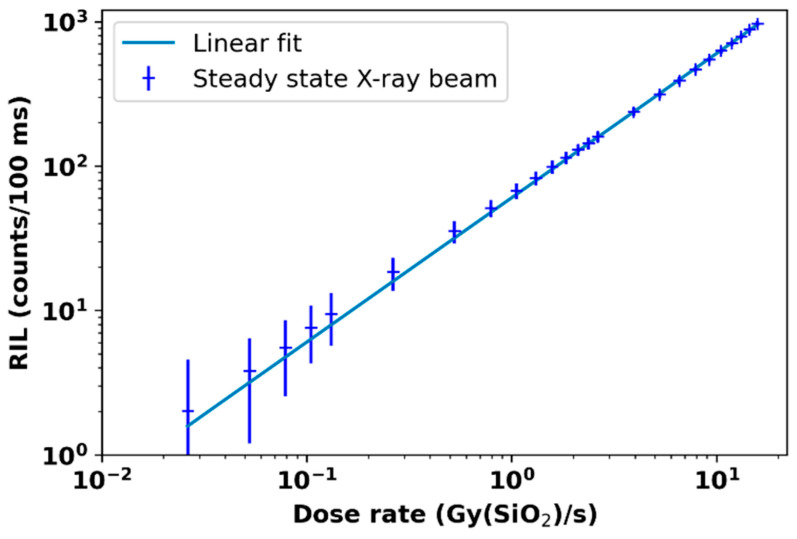
The dose rate dependence of radiation-induced luminescence (RIL) in nitrogen-doped optical fiber was investigated in response to X-ray radiation. Reproduced from reference [65] under the Creative Commons Attribution 4.0 International Li-cense (https://creativecommons.org/licenses/by/4.0/ (accessed on 1 July 2023)).

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
