# Peer review of "FLASH Radiotherapy and the Use of Radiation Dosimeters"

_cancers, 2023, doi:10.3390/cancers15153883_

Round 1

Reviewer 1 Report

FLASH-RT is an important emerging field in medical physics and this paper provides a comprehensive review of the dosimetric challenges involved in its clinical implementation. Although the paper reads like the literature review of an extremely well written PhD thesis, and in that sense lacks original research content, the authors have provided a good service to the medical physics research community with their timely overview. 

The paper is very well written and easy to follow; however, the following minor editorial comments are offered as suggestions for improvements in clarity.

Line 102: The codes EGSnrc/BEAMnrc and DOSXYZnrc MC codes may not be known by all readers, do you have a reference to their origin or where they are available? 

Line 124: What is ‘acoustic imaging’? It is not intuitively obvious what this method of radiation measurement could be. An added sentence or two of how one gets from radiation to sound to an image would be helpful.

Lines 141 to 142: How does the following statement fit into a discussion of FLASH? “Another noteworthy innovation is the pressurized ionization chamber detector, which enables the characterization of alpha and beta radioactive sources and can measure radioactive sources in internal 2π or 4π geometry". 

Line 174: What is an ‘output factor’? 

Lines 322-323: ESRF – European Synchrotron Research Facility?? Please spell out; using rESRF in line 321 first makes it sound like a quantity rather than a quantity at a given facility or research centre. 

Lines 337-339: Figure 4 is problematic. The figure caption does not provide a clear enough description. How are the left and right panels related and what is the relative bias scale on the left-hand panel? 

Reviewer 2 Report

This is a very good review of the status of FLASH radiotherapy concerns. I suggest a change in the title since this paper is essentially a review paper. The title I would suggest is "Review of FLASH Radiotherapy and the use of Radiation Dosimeters." Also I have a few minor comments below.

1.Page 5, line 223 in Figure 2 misspelling - should be ultra-thin,

2 page 6 line 244 the use of" absolute" dose is incorrect. The only absolute dose is one from a primary laboratory such as NIST. Although this is colloquial use, I think we need to start being accurate in our terminology. Film especially can never be used for absolute dosimetry. Just eliminate the word leaving dose.

3. A choice to the authors: would they wish to include a little more in the conclusion on the biological aspects.
